# Soluble Urokinase-Type Plasminogen Activator Receptor (suPAR), Growth Differentiation Factor-15 (GDF-15), and Soluble C5b-9 (sC5b-9) Levels Are Significantly Associated with Endothelial Injury Indices in CAR-T Cell Recipients

**DOI:** 10.3390/ijms252011028

**Published:** 2024-10-14

**Authors:** Eleni Gavriilaki, Christos Demosthenous, Paschalis Evangelidis, Zoi Bousiou, Ioannis Batsis, Anna Vardi, Despina Mallouri, Eudoxia-Evaggelia Koravou, Nikolaos Spyridis, Alkistis Panteliadou, Georgios Karavalakis, Marianna Masmanidou, Tasoula Touloumenidou, Apostolia Papalexandri, Christos Poziopoulos, Evangelia Yannaki, Ioanna Sakellari, Marianna Politou, Ioannis Papassotiriou

**Affiliations:** 1BMT Unit, Hematology Department, George Papanicolaou General Hospital, 57010 Thessaloniki, Greece; christosde@msn.com (C.D.); boussiou_z@hotmail.com (Z.B.); iobats@yahoo.gr (I.B.); anna_vardi@yahoo.com (A.V.); dmallouri@gmail.com (D.M.); evakikor@gmail.com (E.-E.K.); spyridisnik@hotmail.com (N.S.); kirapanteliadou@gmail.com (A.P.); giorgos.karavalakis@gmail.com (G.K.); mariannareti@gmail.com (M.M.); tasoula.touloumenidou@gmail.com (T.T.); lila.papalexandri@gmail.com (A.P.); eyannaki@uw.edu (E.Y.); ioannamarilena@gmail.com (I.S.); 2Second Propedeutic Department of Internal Medicine, Hippocration Hospital, Aristotle University of Thessaloniki, 54642 Thessaloniki, Greece; pascevan@auth.gr; 3Department of Hematology, Metropolitan Hospital, 18547 Athens, Greece; cpozi@otenet.gr; 4Thrombosis–Bleeding–Transfusion Medicine Postgraduate Studies, School of Medicine, National and Kapodistrian University of Athens, 11527 Athens, Greece; mariannapolitou@gmail.com; 5Hematology Laboratory-Blood Bank, Aretaieion Hospital, School of Medicine, National and Kapodistrian University of Athens, 11527 Athens, Greece; 6First Department of Pediatrics, School of Medicine, National and Kapodistrian University of Athens, 11527 Athens, Greece; ipapassotiriou@gmail.com

**Keywords:** chimeric antigen receptor-T (CAR-T), complement, cytokine release syndrome (CRS), endothelial activation and stress index (EASIX), endothelial dysfunction, immune effector cell-associated neurotoxicity syndrome (ICANS), growth differentiation factor-15 (GDF-15), soluble urokinase plasminogen activator receptor (suPAR)

## Abstract

Endothelial injury indices, such as Endothelial Activation and Stress Index (EASIX), modified EASIX (m-EASIX), and simplified EASIX (s-EASIX) scores, have been previously associated with chimeric antigen receptor-T (CAR-T) cell immunotherapy complications. Soluble urokinase-type plasminogen activator receptor (suPAR), growth differentiation factor-15 (GDF-15), and soluble C5b-9 (sC5b-9) have been described as markers of endothelial injury post-hematopoietic stem cell transplantation. In the current study, we examined whether suPAR, GDF-15, and sC5b-9 levels were associated with endothelial injury indices in adult CAR-T cell recipients. The levels of these markers were measured in patients before CAR-T cell infusion and in healthy individuals with immunoenzymatic methods. We studied 45 CAR-T cell recipients and 20 healthy individuals as the control group. SuPAR, GDF-15, and sC5b-9 levels were significantly higher in the patients’ group compared to the healthy control group (*p* < 0.001, in all comparisons). SuPAR levels at baseline were associated with the m-EASIX scores calculated at the same time point (*p* = 0.020), while suPAR and GDF-15 concentrations were correlated with EASIX scores at day 14 post-infusion (*p* < 0.001 in both comparisons). Moreover, sC5b-9 levels were correlated with the s-EASIX scores at infusion (*p* = 0.008) and the EASIX scores at day 14 (*p* = 0.005). In our study, sC5b9, suPAR, and GDF-15 levels were found to reflect endothelial injury in CAR-T cell recipients.

## 1. Introduction

Chimeric antigen receptor-T (CAR-T) immunotherapy constitutes a vital treatment approach for patients with refractory/relapsed B-cell hematological malignancies [1]. Until today, four CAR-T cell products have been approved by the Food and Drug Administration (FDA) and are commercially available for lymphomas and acute lymphoblastic leukemia (ALL): lisocabtagene maraleucel, tisagenlecleucel (tisa-cel), brexucabtagene autoleucel (brexu-cel), and axicabtagene ciloleucel (axi-cel) [2]. Furthermore, recently two CAR-T cell products were approved for the treatment of relapsed/refractory multiple myeloma [3]. Toxicities, including cytokine release syndrome (CRS), immune effector cell-associated neurotoxicity syndrome (ICANS), and cytopenia, are observed in the post-infusion period, increasing the mortality and morbidity that these patients experience [4,5]. Endothelial injury and activation, as a result of both genetic and extrinsic factors’ contributions, have been recognized in the pathogenesis of CRS and ICANS [6]. Moreover, in vitro experimental models have shown that CAR-T cell infusion results in the elevation of endothelial cell inflammatory markers [7]. Endothelial injury indices, such as the Endothelial Activation and Stress Index (EASIX), the modified EASIX (m-EASIX), and the simplified EASIX (s-EASIX) scores, have been associated previously with CAR-T cell-related toxicities, mainly with the onset of severe CRS and ICANS, and reduced overall survival (OS) [8,9,10]. The EASIX score was initially developed for hematopoietic stem cell transplantation (HSCT) recipients; however, associations have been found not only between EASIX scores and mortality in these patients, but also with the development of HSCT-related complications, including HSCT-associated thrombotic microangiopathy (HSCT-TMA) and graft-versus-host disease [11,12,13].

Increased activation of the complement system cascade has been recognized in various hematological (such as sickle cell disease) and non-hematological conditions [14]. The complement system can be activated via three pathways: classical, lectin, and alternative [15]. Thus, the terminal pathway is activated, leading to the formulation of the membrane attack complex (MAC), which comprises a complement C5b-9 component, and subsequently results in cell lysis. Complement system activation has been found to be a major contributor to the pathophysiology of HSCT-TMA and other endothelial injury syndromes post-HSCT [16]. Moreover, sC5b-9 levels have been associated with HSCT-TMA onset and severity, while complement inhibition constitutes the main treatment approach for these patients [17,18]. Complement activation has never been studied in CAR-T cell recipients.

Soluble urokinase plasminogen activator receptor (suPAR) is a blood protein that initially had been described as a marker of systemic chronic inflammation in patients with chronic kidney disease [19]. Moreover, growth differentiation factor-15 (GDF-15) is expressed by human cells under circumstances of oxidative stress, aging, and inflammation [20,21]. Increased GDF-15 expression has been demonstrated in various clinical entities, including hematological and non-hematological disorders, such as cardiovascular and metabolic diseases [20]. SuPAR and GDF-15 levels have been measured in allogeneic HSCT (allo-HSCT) recipients and correlated with the development of HSCT-associated endothelial injury syndromes, such as GVHD and HSCT-TMA, and endothelial injury indices [22,23]. The potential role of suPAR and GDF-15 as markers of endothelial activation and injury in CAR-T cell recipients remains unknown.

Our group has found associations between these markers and EASIX scores calculated during the follow-up period of HSCT recipients; additionally, a correlation was found between suPAR levels and sC5b-9 levels, which serve as diagnostic and prognostic markers of HSCT-TMA in which endothelial dysfunction can be observed [22,24]. Given the common pathophysiology behind endothelial injury syndromes in allo-HSCT and post-CAR-T cell immunotherapy, and the immune dysregulation during CRS and ICANS, we speculated that well-established markers of endothelial activation in allo-HSCT recipients, such as sC5b-9, might also reflect endothelial injury in these patients [6].

## 2. Results

### 2.1. Patients Characteristics

Forty-five patients were included in the study, with a median age of 47 years old (range: 21–75). The patients’ conditions indicated that CAR-T cell therapy was needed: diffuse large B cell lymphoma (DLBCL) in 30 (66.7%) patients, primary mediastinal large B-cell lymphoma (PMBCL) in three (6.7%), ALL in six (13.3%), and mantle cell lymphoma (MCL) in six (13.3%). Axi-cel was infused in 24 patients (53.3%), tisa-cel in 14 (31.1%), and brexu-cel in seven (15.6%). The median line of previous treatment was three (2–9). Three patients with DLBCL had received a previous autologous HSCT, while allo-HSCT had been performed in four patients, three with ALL and one with DLBCL. Thirty-five (77.7%) patients received bridging chemotherapy before the lymphodepleting therapy. The clinical and laboratory characteristics (assessed before their admission to the Cellular Therapy Unit, on the same day of sC5b-9, suPAR, and GDF-15 samplings) of the study participants are presented in Table 1.

### 2.2. sC5b-9, SuPAR and GDF-15 Levels

SC5b-9 levels (median: 209.9 ng/mL, range: 99.9–749.7) were significantly increased in patients who received CAR-T cell therapy in comparison to the healthy control group (*p* < 0.001). The levels of suPAR (median: 3.7 ng/mL; range: 2–31.2) were significantly higher in CAR-T cell recipients compared to the healthy control group (*p* < 0.001). Similarly, GDF-15 levels (median: 2807.5, range: 569.2–8655) in patients who received CAR-T cell therapy were significantly increased in comparison to the healthy control group (*p* < 0.001). We did not identify any correlations between the levels of su-PAR, GDF-15, and s-C5b9 and the factors as follows: age, disease entity, disease phase, CAR-T cell product infused, median number of previous lines of therapy, and previous HSCT. Then, we further analyzed the characteristics of CAR-T cell recipients.

### 2.3. Univariate Associations of sC5b-9, suPAR, and GDF-15 Levels with EASIX Indices and Laboratory Values

Correlations were identified between sC5b-9 levels at baseline, s-EASIX scores at infusion (r = 0.468, *p* = 0.008), and EASIX14 scores (r = 0.496, *p* = 0.005). These data are presented in Figure 1. Moreover, sC5b-9 values were associated with LDH levels at any time point (baseline: r = 0.374, *p* = 0.038, infusion r = 0.386, *p* = 0.032, day 14 r = 0.397, *p* = 0.027).

SuPAR levels at baseline were associated with the m-EASIX scores, calculated at the same time point (r = 0.349, *p* = 0.020). SuPAR and GDF-15 concentrations were associated with EASIX scores at day 14 post-infusion (r = 0.885, *p* < 0.001, and r = 0.557, *p* < 0.001, respectively). Among the laboratory values used to calculate EASIX scores (LDHs, creatinines, and platelets), suPAR levels were associated with LDH levels during infusion (r = 0.312, *p* = 0.039) and platelets at day 14 (r = 0.295, *p* = 0.05). Furthermore, GDF-15 levels were associated with ferritin levels at baseline (r = -0.305, *p* = 0.044). These data are summarized in Table 2.

### 2.4. Multivariate Analysis

Linear regression analysis was performed in the study population for parameters that were significantly associated with suPAR and GDF-15 levels in the univariate analysis. In the multivariate regression analysis, suPAR levels were correlated with m-EASIX scores at baseline (*p* < 0.001), EASIX scores at day 14 (*p* = 0.017), and LDH values at infusion (*p* < 0.001), while GDF-15 levels were associated with baseline ferritin values (*p* = 0.036) and EASIX scores at day 14 (*p* < 0.001).

### 2.5. CRS and ICANS

Severe CRSs (grade ≥ 3) developed in six CAR-T cell recipients (13.3%), while severe ICANSs (grade ≥ 3) were observed in three (6.6%). Tocilizumab was administered to 36 patients, while corticosteroids in 23. SC5b-9, SuPAR, and GDF-15 levels at baseline did not differ between patients who developed severe CRS and/ or ICANS and those who did not (*p* = 0.289, *p* = 0.330, *p* = 0.927, respectively). The levels of the biomarkers did not correlate with other studied factors.

### 2.6. Overall and Progression Free Survival

With a median follow-up of 10.5 (1–42) months, OS and progression-free survival (PFS) rates at 1 year were 80% and 37.8%, respectively. Serum levels of sC5b-9 (*p* = 0.023) and LDH at baseline (*p* = 0.033), EASIX (*p* = 0.023) and s-EASIX (*p* = 0.039) scores at infusion, and s-EASIX14 scores (*p* = 0.016) were correlated with OS. Among these factors, only the s-EASIX scores calculated on the day of the infusion were independent predictors of overall survival (β = 5.661, *p* = 0.012) in the multivariate analysis. PFS was not correlated with any of the studied factors.

S-EASIX0 (area under the curve (AUC): 0.728, *p* = 0.018) and s-EASIX14 (AUC: 0.695, *p* = 0.043) scores predicted the risk of death in the receiver operating characteristic (ROC) curves, as shown in Appendix A). Moreover, the s-EASIX0 score over the median value (1.9, range: 0.74–49.3) was associated with poor OS (*p* = 0.027), similarly to the s-EASIX14 score over the median value (3, range: 0.6–74.6), (*p* = 0.004). These data are presented in Figure 2. SC5b-9 levels at baseline predicted the risk of death 12 months post-infusion in the ROC curve (AUC: 0.797, *p* = 0.026), as shown in the Appendix A).

## 3. Discussion

In this real-world study, for the first time, we investigated the potential associations between suPAR and GDF-15 markers and endothelial injury indices in patients who were going to receive CAR-T cell immunotherapy. Moreover, we showed that these markers were significantly higher in patients who received CAR-T cell therapy compared to a healthy control group. SuPAR levels were associated with the m-EASIX scores calculated at the same time point and EASIX14 scores, while GDF-15 values were correlated with EASIX14 scores. Among the laboratory values used for the calculation of EASIX scores on day 14, a correlation was found between suPAR levels and platelet counts, suggesting that the association with EASIX is not driven by laboratory values alone. Furthermore, levels of sC5b-9 were measured in our study population and were found significantly increased in comparison to the healthy control group. An association was shown between sC5–9 values assessed before the infusion of the CAR-T product and endothelial injury indices (s-EASIX at infusion and EASIX14 scores). For the first time, s-EASIX scores calculated at the day of the infusion and early in the post-therapy period were found to be associated with OS.

Luft and associated colleagues developed and validated the EASIX score not only as a marker of endothelial injury in allo-HSCT recipients, finding associations between this score and HSCT-TMA onset, but also established the EASIX score as a predictor of mortality [12]. Moreover, in their work, correlations had been reported between EASIX scores and biomarkers of endothelial activation, such as interleukin-6 (IL-6), interleukin-18, and insulin-like-growth-factor-1 [11]. Our group has shown the potential relationship between EASIX scores, markers of endothelial injury, and complement activation, found a significant correlation between EASIX scores, and calculated 100 post allo-HSCT and soluble C5b-9 (sC5b-9) levels [13]. SC5b-9 values above the normal limit are included in the novel diagnostic criteria of HSCT-TMA [25]. HSCT-TMA development can be attributed to genetic factors, endothelial activation, and complement activation, leading to endothelial injury and the activation of the coagulation cascade [26,27,28]. Furthermore, future research directions include the investigation of genetic factors influencing complement activation in these patients.

Endothelial cells’ activation has been suggested as a crucial element in the pathogenesis of CRS and ICANS [29]. Biomarkers of endothelial activation, such as angiopoietin 2 and von Willebrand Factor (vWF), have been found increased in patients who received CAR-T cell therapy and developed very severe neurotoxicity (ICANS grade ≥ 4) [30]. A lower ADAMTS13/vWF ratio in patients was also reported in comparison to patients with lower-grade ICANS [30]. ADAMTS13 (a disintegrin and metalloproteinase with a thrombospondin type 1 motif, member 13) is a metalloproteinase that cleaves the polymers of vWF, and diminished activity of ADAMTS13 is the diagnostic hallmark of thrombotic thrombocytopenic purpura. Levels of intercellular adhesion molecule 1 (ICAM-1), vascular cell adhesion protein 1 (VCAM-1), and of other endothelial injury markers have been related to the CRS pathogenesis [31]. In an experimental study, therapeutic blockade of interleukin 1 beta (IL-1β) and tumor necrosis alpha (TNF-a), which are major cytokines contributing to the endothelial damage post-CAR-T infusion and CRS/ICANS pathogenesis, have been found to protect against endothelial damage [31]. Thus, CRS and ICANS can be considered as “endotheliopathies”. However, these experimental data have to be validated in real-world CAR-T cell recipients, and blood sampling for the measurement of endothelial cell activation biomarkers should be performed both before the onset of CRS/ICANS and during the complications. In the current study, the levels of suPAR and GDF-15 measured before the lymphodepleting therapy administration did not differ among the patients who developed severe CRS and/or ICANS post the infusion; in ROC analysis, these markers failed to predict the onset of severe complications. This might be attributed to the multifactorial causes implicated in the pathogenesis of CRS and ICANS, which include not only endothelial injury but also immune dysregulation. In the future, we plan to measure the suPAR and GDF-15 values during the post-infusion period at several time points to find possible associations with severe ICANS and CRS development. More data regarding the clinical significance of these biomarkers are essential in order to incorporate them in everyday clinical decision-making.

Based on the role of EASIX scores in the prediction of endothelial injury syndromes post-allo-HSCT and the endothelial activation that is observed during CRS and ICANS post-CAR-T cell infusion, researchers have investigated in various studies the potential prognostic role of EASIX scores in CAR-T-associated complications and mortality. Korell et al. in their retrospective study showed the predictive role of EASIX scores calculated before lymphodepleting therapy administration for the development of Grade ≥ 3 CRS and ICANS [32]. Furthermore, they showed significant associations between this score and markers of endothelial activation, such as angiopoietin-2, suppressor of tumorigenicity 2, soluble thrombomodulin, and interleukin 8, measured at the same time point [32]. De Boer and associated colleagues found that EASIX, m-EASIX, and s-EASIX scores estimated before lymphodepleting therapy can be useful in the identification of high-risk patients for ICANS grade ≥ 2. In the study by Pennisi et al., m-EASIX scores calculated the day before the CAR-T infusion and in the early post-infusion period (days 1 and 3) were found to be predictive for CRS, neurotoxicity, and disease response [8]. Recently, all three EASIX scores have been associated with the development of severe CRS and ICANS in patients who received C-type lectin-like molecule 1 CAR-T cell immunotherapy for acute myeloid leukemia [33]. In a previous paper from our group, we were the first to show that EASIX and m-EASIX scores calculated (m-EASIX14) at 14 days post-infusion were associated with OS in CAR-T cell recipients and that m-EASIX14 scores were also correlated with PFS [10]. Interestingly, endothelial activation indices have been correlated with increased mortality in patients with severe COVID-19 [34]. It is considered crucial to examine the possible associations between endothelial activation indices estimated before the infusion, during the early post-infusion period, and at the follow-up with other CAR-T cell-related complications, such as hematological toxicity, infections, relapses, and secondary malignancies.

SuPAR can be identified in blood and other blood fluids and is the soluble form of the uPAR receptor [19]. SuPAR is released in the circulation from proteolytic cleavage of uPAR, which is found in cells’ membranes, during inflammatory procedures [35]. SuPAR levels have been associated with several inflammatory markers, such as C-reactive protein (CRP), TNF-a, and IL-6, in various clinical entities [36,37,38]. However, in our analysis, a correlation was not identified between suPAR and CRP values. This marker has widely been associated with chronic kidney disease severity and several other inflammatory, infectious, and cardiovascular (CVD) diseases [19,39]. GDF-15 is also expressed in the tissues under circumstances of hypoxia, inflammation, and oxidative stress [14]. Similarly to suPAR, GDF-15 has been proposed as a marker of CVD [40]. Patients who receive allo-HSCT and other cellular therapies experience an increased burden of CVD, as has been shown by clinical and laboratory data [41,42,43]. Correlations between the suPAR/GDF-15 levels assessed before the administration of therapy (allo-HSCT or CAR-T cell product), during follow-up, and when burdened by CVD in those patient groups should be investigated.

In our study, several limitations should be acknowledged. First, this is a single-center study in which only CAR-T cell recipients from our center were enrolled, and a limited number of patients participated. Moreover, long-term follow-up would be useful to find potential correlations between the proposed markers and other CAR-T cell-related complications. Multi-center studies, including a large amount of CAR-T cell recipients, are essential in order to validate our data.

## 4. Materials and Methods

### 4.1. Cohort and Study Design

This is an observational study. The main aim of our study was to examine the possible associations between sC5b-9, suPAR, and GDF-15 levels and EASIX scores in CAR-T cell recipients. Furthermore, we investigated the potential correlation among these markers, severe CRS and ICANS development, OS, and PFS. We consecutively enrolled adult patients (≥18 years old) with relapsed/refractory lymphomas or B-ALL who received commercially available CAR-T cell products, according to the current guidelines at the JACIE-accredited Unit of the Hematology Clinic of G. Papanicolaou Hospital, in Thessaloniki, Greece. The control group composed of healthy adult individuals of similar age and gender. Samples of the patients used for the measurement of sC5b-9, suPAR, and GDF-15 levels were collected before their admission to the Cellular Therapy Unit, centrifuged, rapidly aliquoted, and stored immediately at −80 °C. The samples from consecutive healthy volunteers were collected at the outpatient department during arranged appointments and stored after following the previously described procedure.

All patients received lymphodepleting therapy before CAR-T cell infusion, with cyclophosphamide and fludarabine, according to each products’ protocol [44]. Tisa-cel and axi-cel were administered since 2020 and brexu-cel since 2022. For patient monitoring, as well as for prophylaxis and treatment of side effects, the EBMT and MD Anderson guidelines were adopted [44,45]. CRS and ICANS were diagnosed and graded according to the American Society for Transplantation and Cellular Therapy (ASTCT) grading system [4]. CRS and ICANS grade ≥3 were considered severe. Levetiracetam prophylactic treatment was administered to the majority of the CAR-T cell recipients from the day of the infusion [46,47]. The data cut-off date was the end of June 2024, with at least 1 month of follow-up post-infusion. The essential clinical and laboratory data were extracted from the patients’ medical records. The study was approved by the institutional review board of G. Papanikolaou Hospital, and all patients included in the study provided written informed consent in adherence to the principles of the Helsinki Declaration.

### 4.2. EASIX, s-EASIX, m-EASIX Calculation and sC5b-9, suPAR, GDF-15 Measurement

Laboratory values of the patients were retrieved from patients’ electronic medical records for the calculation of EASIX, s-EASIX, and m-EASIX scores at different time points. EASIX [lactate dehydrogenase (U/L) × creatinine (mg/dL)/thrombocytes (10^9^ cells per L)] and s-EASIX scores were calculated at baseline (EASIXbaseline and s-EASIXbaseline are the scores before admission to the Unit and lymphodepleting therapy administration), at the day of CAR-T cell product infusion (EASIX0, s-EASIX0), and 14 days post CAR-T infusion (EASIX14, s-EASIX14). In the calculation of s-EASIX scores, creatinine values were excluded. M-EASIX scores, in which the creatinine values are replaced by the CRP values, were calculated at baseline (m-EASIXbaseline). The following formulas were used for the calculation of EASIX scores:EASIX: [lactate dehydrogenase (U/L) × creatinine (mg/dL)]/thrombocytes (10^9^ cells per L)m-EASIX: [lactate dehydrogenase (U/L) × CRP (mg/dL)]/thrombocytes (10^9^ cells per L)s-EASIX: lactate dehydrogenase (U/L)/thrombocytes (10^9^ cells per L)

Immunoenzymatic techniques were used for the measurement of sC5b-9/MAC (Quidel, San Diego, CA, USA), suPAR (ViroGates A/S, Birkerod, Denmark), and GDF-15 levels (R&D Systems, Minneapolis, MN, USA). In Figure 3, we present the methodology followed in our work.

### 4.3. Statistical Analysis

SPSS 28.0 (IBM SPSS Statistics for Windows, Version 28.0. Armonk, NY, USA: IBM Corp) was used for the statistical analysis of the data. Categorial variables were presented as frequencies. The following factors were analyzed: age; disease; CAR-T cell product infused; prior treatment lines; prior history of HSCT, EASIX, m-EASIX, and s-EASIX scores; laboratory values (lactate dehydrogenases, creatinines, platelets, CRPs, ferritins); sC5b-9, suPAR, and GDF-15 levels; severe CRS and ICANS development; and overall survival and progression-free survival. Quantitative variables were tested with the Shapiro–Wilk procedure for the determination of the normality of distribution, while non-normal variables were expressed as median, range, and normal as mean. The Mann–Whitney test for comparison of nonparametric variables was performed between the groups. Logarithmic transformation of a non-to-normal distribution was performed when considered necessary. Spearman’s rank correlation coefficients were used to characterize the bivariate correlations. A logistic regression model was used for multivariate analysis. A ROC curve was performed for the analysis of sensitivity, specificity, and cutoff value. For survival analysis, the Kaplan–Meier method was used, while long-rank test was performed for the comparisons between the different groups. Multivariate analysis for survival was performed using a Cox regression model and the “enter” method. The statistical significance level was set at *p* < 0.05.

## 5. Conclusions

In this observational study, we examined the possible correlations between sC5b-9, suPAR, and GDF-15 levels and endothelial injury indices in 45 CAR-T cell recipients. SC5b-9, SuPAR and GDF-15 levels were significantly higher in patients in comparison to healthy control group (*p* < 0.001, in all comparisons). We identified correlations between suPAR levels, GDF-15 levels, and EASIX14 scores, while m-EASIX scores at baseline were associated only with suPAR levels. SC5b-9 values also correlated with endothelial injury indices calculated at different time points. Furthermore, these biomarkers correlated with laboratory values assessed pre- and post-infusion. Our study shows for the first time that sC5b-9, suPAR, and GDF-15 reflect endothelial injury in CAR-T cell recipients. In accordance with other patient populations, suPAR and GDF-15 emerge as markers of endothelial dysfunction, characterizing high-risk in endothelial injury syndromes, and, in particular, CRS. As has been shown before by our group, endothelial injury indices serve as useful markers for prediction of survival in CAR-T cell recipients. However, before their clinical usefulness can be demonstrated, rigorous validation in multiple cohorts is warranted. In the era of personalized medicine, early recognition and efficient prophylactic interventions of CAR-T cell-related toxicities are considered essential.

## Figures and Tables

**Figure 1 ijms-25-11028-f001:**
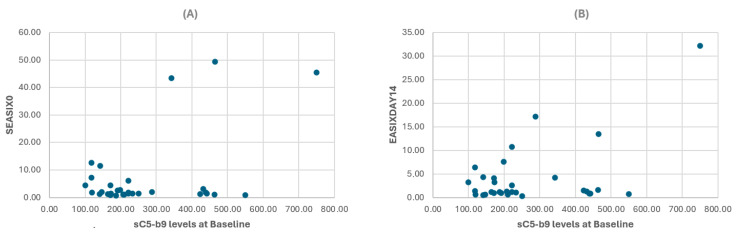
(**A**) sC5b9 levels show a significant correlation with sEASIX0 (*p* = 0.008) and (**B**) EASIX14 (*p* = 0.005) scores. sC5b9 = soluble C5b-9 and s-EASIX0 = simplified Endothelial Activation and Stress Index scores were calculated at the day of CAR-T cell infusion; EASIXDAY14 = Endothelial Activation and Stress Index scores were calculated 14 days post-CAR-T cell infusion.

**Figure 2 ijms-25-11028-f002:**
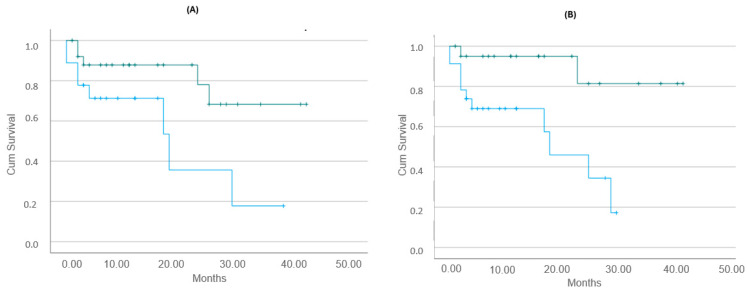
(**A**) s-EASIX0 score over the median value (1.9, range: 0.74–49.3) was associated with poor OS (*p* = 0.027); (**B**) s-EASIX14 score over the median value (3, range: 0.6–74.6) was also associated with poor OS (*p* = 0.004). Green lines: s-EASIX score below the median; blue lines: s-EASIX score over the median value. S-EASIX0 = simplified Endothelial Activation and Stress Index calculated at the day of the infusion; S-EASIX14 = simplified Endothelial Activation and Stress Index calculated 14 days post-infusion; OS = overall survival.

**Figure 3 ijms-25-11028-f003:**
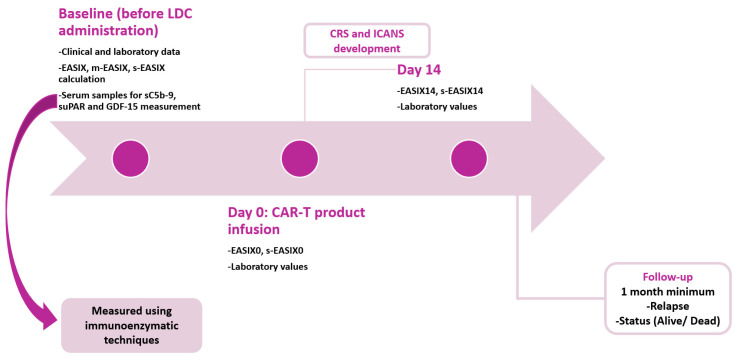
Methodology used in our study. Samples for sC5b-9, suPAR, and GDF-15 measurements were obtained from CAR-T cell recipients before their admission to the cellular therapy unit, and at the same time point, EASIX, mEASIX, and s-EASIX scores were calculated. EASIX and s-EASIX scores were also calculated on the day of CAR-T cell product infusion and 14 days post-infusion. During the post-infusion period, patients were closely monitored for therapy-related toxicities. The minimum follow-up period was 1 month. LDC = lymphodepleting chemotherapy; EASIX = Endothelial Activation and Stress Index; m-EASIX = modified Endothelial Activation and Stress Index; s-EASIX = simplified Endothelial Activation and Stress Index; sC5b-9 = soluble C5b-9; suPAR = soluble urokinase-type plasminogen activator receptor; GDF-15 = growth differentiation factor-15; CRS = cytokine release syndrome; ICANS = immune effector cell-associated neurotoxicity syndrome.

**Table 1 ijms-25-11028-t001:** Clinical and laboratory characteristics of the study participants. Laboratory values were obtained before the admission to the Cellular Therapy Unit and LDC administration, on the same day of sC5b-9, suPAR, and GDF-15 samplings.

Baseline Characteristics	Patients (*N* = 45)
Median age, years (range)	47 (21–75)
Disease entity, *n* (%)	
DLBCL	30 (66.7%)
PMCL	3 (6.7%)
ALL	6 (13.3%)
MCL	6 (13.3%)
Disease status, *n* (%)	
Refractory	37 (82.2%)
2nd relapse	6 (13.3%)
3rd relapse	2 (4.4%)
CAR-T product, *n* (%)	
Axicabtagene ciloleucel	24 (53.3%)
Tisagenlecleucel	14 (31.1%)
Brexucabtagene autoleucel	7 (15.6%)
Median no. of previous lines of therapy (range)	3 (2–9)
Previous HSCT, *n* (%)	
Auto-HSCT	3 (6.7%)
Allo-HSCT	4 (8.9%)
Bridging therapy, *n* (%)	35 (77.7%)
Markers before LDC	
Median C-reactive protein (mg/dL), range	0.75 (0.4–7.49)
Median ferritin (ng/mL), range	336 (4.8–7928)
Median serum LDH (U/L), range	266 (135–1411)
Mean platelets (×10^9^/L), 95% CI	144.8 (122.8, 166.9)
Median creatinine (mg/dL), range	0.84 (0.39–1.85)
Median EASIX score before LDC, range	1.9 (0.4–22.4)
Median m-EASIX score before LDC, range	1.6 (0.08–163.4)
Median s-EASIX score before LDC, range	2.1 (0.57–25.1)

LDC = lymphodepleting therapy; sC5b-9 = soluble C5b-9; suPAR = soluble urokinase-type plasminogen activator receptor; GDF-15 = growth differentiation factor-15; DLBCL = diffuse large B cell lymphoma; PMCL = primary mediastinal large B-cell lymphoma; ALL = acute lymphoblastic leukemia; MCL = mantle cell lymphoma; CAR-T = chimeric antigen receptor-T; HSCT = hematopoietic stem cell transplantation; Auto-HSCT = autologous hematopoietic stem cell transplantation; Allo-HSCT = allogeneic hematopoietic stem cell transplantation; LDH = lactate dehydrogenase; CI = confidence interval; EASIX = Endothelial Activation and Stress Index; m-EASIX = modified Endothelial Activation and Stress Index; s-EASIX = simplified Endothelial Activation and Stress Index.

**Table 2 ijms-25-11028-t002:** Univariate Associations of sC5b-9, suPAR and GDF-15 levels with EASIX indices and laboratory values.

Variable	Correlation Analysis for sC5b-9		Correlation Analysis for suPAR		Correlation Analysis for GDF-15	
	Correlation Coefficient r	*p* Value	Correlation Coefficient r	*p* Value	Correlation Coefficient r	*p* Value
s-EASIX at infusion	0.468	0.008	-	-		
m-EASIX score at baseline	-	-	0.349	0.020	-	-
EASIX score at day 14 post-infusion	0.496	0.005	0.885	<0.001	0.557	<0.001
Ferritin levels at baseline	-	-	-	-	-0.305	0.044
LDH at baseline	0.374	0.038	-	-		
LDH levels during infusion	0.386	0.032	0.312	0.039	-	-
LDH at day 14	0.397	0.027	-	-	-	-
Platelets at day 14	-	-	0.295	0.05	-	-

sC5b-9 = soluble C5b-9; s-EASIX = suPAR = soluble urokinase-type plasminogen activator receptor; GDF-15 = growth differentiation factor-15; s-EASIX = simplified Endothelial Activation and Stress Index; EASIX = Endothelial Activation and Stress Index; m-EASIX = modified Endothelial Activation and Stress Index; LDH = lactate dehydrogenase.

## Data Availability

Data will be readily available upon request to the corresponding author.

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
