# Peer review of "Soluble Urokinase-Type Plasminogen Activator Receptor (suPAR), Growth Differentiation Factor-15 (GDF-15), and Soluble C5b-9 (sC5b-9) Levels Are Significantly Associated with Endothelial Injury Indices in CAR-T Cell Recipients"

_ijms, 2024, doi:10.3390/ijms252011028_

Round 1
Reviewer 1 Report
Comments and Suggestions for Authors
This simple and well conducted clinically relevant study tests the impact of Chimeric antigen receptor-T (CAR-T) therapy on endothelial well being by assessing well established markers (soluble urokinase plasminogen activator receptor, suPAR; growth differentiation factor-15, GDF-15; soluble C5b-9, sC5b-9) for endothelial injury in subjects receiving CAR-T therapy. The authors demonstrate a positive association between the established endothelial activation and stress index (EASIX) and su-PAR, GDF-15 and sC5b-9 in subjects receiving CAR-T. The study design, methodology and data analysis is appropriate and well established to address their goals. The paper is well written an easy to follow and the findings clinically relevant. The conclusions accurately reflect their findings. I have a few suggestions that may be help further improve the paper.
1: What are the markers used for EASIX assessment in clinics? they should be better defined.
2: Is the impact of CAR-T on endothelial health direct or indirect ? Is there data to show that endothelial cells respond CAR-T in vitro ? If not - it would be a simple approach to test the effects of CAR-T in cultured human vascular ECs. It would be helpful if the authors could look at the levels of EC markers in response to CAR-T in cultured ECs. This would strengthen their clinical findings.
Author Response
Reviewer 1
This simple and well conducted clinically relevant study tests the impact of Chimeric antigen receptor-T (CAR-T) therapy on endothelial well being by assessing well established markers (soluble urokinase plasminogen activator receptor, suPAR; growth differentiation factor-15, GDF-15; soluble C5b-9, sC5b-9) for endothelial injury in subjects receiving CAR-T therapy. The authors demonstrate a positive association between the established endothelial activation and stress index (EASIX) and su-PAR, GDF-15 and sC5b-9 in subjects receiving CAR-T. The study design, methodology and data analysis is appropriate and well established to address their goals. The paper is well written an easy to follow and the findings clinically relevant. The conclusions accurately reflect their findings.
Answer: We are deeply thankful for the positive feedback. Moreover, we would like to thank the reviewer for the time dedicated to reviewing our manuscript. Indeed, the suggestions of the reviewer were valuable for the quality of our work.
I have a few suggestions that may be help further improve the paper.
1: What are the markers used for EASIX assessment in clinics? they should be better defined.
Answer: Thanks for this comment! The formulas used for the calculation of EASIX scores were included in the revised version of the manuscript (4. Methods: 4.2 EASIX, s-EASIX, m-EASIX calculation and sC5b-9, suPAR, GDF-15 measurement)
2: Is the impact of CAR-T on endothelial health direct or indirect ? Is there data to show that endothelial cells respond CAR-T in vitro ? If not - it would be a simple approach to test the effects of CAR-T in cultured human vascular ECs. It would be helpful if the authors could look at the levels of EC markers in response to CAR-T in cultured ECs. This would strengthen their clinical findings.
Answer: We would like to thank you for this interesting idea! In vitro experimental models have shown that CAR-T cell infusion results in the elevation of endothelial cell inflammatory markers. We added these data in our revised manuscript.
Reviewer 2 Report
Comments and Suggestions for Authors
Title: Soluble Urokinase-Type Plasminogen Activator Receptor (suPAR), Growth Differentiation Factor-15 (GDF-15), and Soluble C5b-9 (sC5b-9) Levels Are Significantly Associated with Endothelial Injury Indices in CAR-T Cell Recipients
This manuscript makes a significant contribution to the expanding literature on CAR-T cell therapy and endothelial injury. The findings are novel, well-supported by data, and presented clearly. However, the study could be enhanced by addressing a few limitations, such as the small sample size, providing a more comprehensive discussion of non-significant findings, and including more details on control group variances and potential confounding variables. Future research directions are promising, especially in exploring the role of complement activation in CAR-T-related toxicities. Below are my suggestions to enhance the manuscript:
Line 79:
The introduction provides a solid overview of complement activation in hematopoietic stem transplantation (HSCT). However, it would benefit from a more detailed discussion on its potential role in CAR-T therapy. Please consider including an explanation, along with citations, regarding why complement activation might be relevant to CAR-T patients.
Line 103:
While a healthy control group is included, the paper does not address whether other factors, such as comorbidities or prior treatments like chemotherapy, may have influenced biomarker levels. Providing more information about the screening process and treatments in the control group would improve comparisons and help address potential confounding issues.
Line 171-173:
The results indicate that suPAR and GDF-15 levels did not predict severe CRS or ICANS. However, a more detailed explanation for these non-significant findings would be helpful. Including possible reasons why these markers failed to predict these outcomes could enhance the discussion.
Line 246:
While the discussion mentions plans to measure suPAR and GDF-15 at later time points, proposing more specific future research directions would strengthen the paper. For instance, exploring genetic factors influencing complement activation or studying these biomarkers in a larger CAR-T patient population with longer follow-up could be beneficial.
Line 239:
Given the relevance of these biomarkers in predicting CAR-T-related toxicities, a brief discussion on how these findings might eventually influence clinical decision-making would be valuable. Focusing on how these biomarkers could be used to manage high-risk CAR-T patients would enhance the practical application of the research.
Line 289:
The discussion acknowledges the small sample size and single-center nature of the study, but expanding on how these limitations may affect the generalizability of the results would be helpful. It would also be beneficial to recommend strategies for future multicenter studies to address these issues.
Line 383:
The conclusion could be improved by further exploring the clinical significance of these findings. Explaining how these biomarkers can be utilized in monitoring or treating CAR-T patients would highlight the practical applications of this research.
Author Response
Reviewer 2
Title: Soluble Urokinase-Type Plasminogen Activator Receptor (suPAR), Growth Differentiation Factor-15 (GDF-15), and Soluble C5b-9 (sC5b-9) Levels Are Significantly Associated with Endothelial Injury Indices in CAR-T Cell Recipients
This manuscript makes a significant contribution to the expanding literature on CAR-T cell therapy and endothelial injury. The findings are novel, well-supported by data, and presented clearly.
Answer: We thank the reviewer for the time dedicated to reviewing our manuscript. Furthermore, we are grateful for the valuable comments and suggestions.
However, the study could be enhanced by addressing a few limitations, such as the small sample size, providing a more comprehensive discussion of non-significant findings, and including more details on control group variances and potential confounding variables. Future research directions are promising, especially in exploring the role of complement activation in CAR-T-related toxicities. Below are my suggestions to enhance the manuscript:
Line 79:
The introduction provides a solid overview of complement activation in hematopoietic stem transplantation (HSCT). However, it would benefit from a more detailed discussion on its potential role in CAR-T therapy. Please consider including an explanation, along with citations, regarding why complement activation might be relevant to CAR-T patients.
Answer: We would like to thank the reviewer for this interesting idea. Given the common pathophysiology behind endothelial injury syndromes in allo-HSCT and endothelial injury post-CAR-T cell immunotherapy, and the immune dysregulation during CRS and ICANS, we speculated that well-established markers of endothelial activation in allo-HSCT recipients, such as sC5b-9 might reflect endothelial injury also in these patients.
Line 103:
While a healthy control group is included, the paper does not address whether other factors, such as comorbidities or prior treatments like chemotherapy, may have influenced biomarker levels. Providing more information about the screening process and treatments in the control group would improve comparisons and help address potential confounding issues.
Answer: The reviewer is right. We did not identify any correlations between levels of su-PAR, GDF-15, and s-C5b9 and age, disease entity, disease phase, CAR-T cell product infused, median number of previous lines of therapy, and previous HSCT. Moreover, the control group comprised of healthy adult individuals of similar age and gender.
Line 171-173:
The results indicate that suPAR and GDF-15 levels did not predict severe CRS or ICANS. However, a more detailed explanation for these non-significant findings would be helpful. Including possible reasons why these markers failed to predict these outcomes could enhance the discussion.
Answer: Thanks for this suggestion. This can be attributed to the multifactorial causes implicated in the pathogenesis of CRS and ICANS, which include not only endothelial injury but also immune dysregulation.
Line 246:
While the discussion mentions plans to measure suPAR and GDF-15 at later time points, proposing more specific future research directions would strengthen the paper. For instance, exploring genetic factors influencing complement activation or studying these biomarkers in a larger CAR-T patient population with longer follow-up could be beneficial.
Answer: We are grateful for this idea. The discussion section was updated accordingly.
Line 239:
Given the relevance of these biomarkers in predicting CAR-T-related toxicities, a brief discussion on how these findings might eventually influence clinical decision-making would be valuable. Focusing on how these biomarkers could be used to manage high-risk CAR-T patients would enhance the practical application of the research.
Answer: Thanks for pointing this out! We made the suggested changes in the revised version of our work.
Line 289:
The discussion acknowledges the small sample size and single-center nature of the study, but expanding on how these limitations may affect the generalizability of the results would be helpful. It would also be beneficial to recommend strategies for future multicenter studies to address these issues.
Answer: The reviewer is right. The manuscript was updated accordingly.
Line 383:
The conclusion could be improved by further exploring the clinical significance of these findings. Explaining how these biomarkers can be utilized in monitoring or treating CAR-T patients would highlight the practical applications of this research.
Answer: Thanks for this idea! We have updated the conclusions section accordingly.